# A Gut Reaction to SIV and SHIV Infection: Lower Dysregulation of Mucosal T Cells during Acute Infection Is Associated with Greater Viral Suppression during cART

**DOI:** 10.3390/v13081609

**Published:** 2021-08-14

**Authors:** Megan A. O’Connor, Paul V. Munson, Sandra E. Dross, Hillary C. Tunggal, Thomas B. Lewis, Jessica Osborn, Christopher W. Peterson, Meei-Li W. Huang, Cassandra Moats, Jeremy Smedley, Keith R. Jerome, Hans-Peter Kiem, Kenneth C. Bagley, James I. Mullins, Deborah Heydenburg Fuller

**Affiliations:** 1Department of Microbiology, University of Washington, 750 Republican St., Seattle, WA 98109, USA; meganoc@uw.edu (M.A.O.); pmunson@parkerici.org (P.V.M.); sdross@uw.edu (S.E.D.); htunggal@uw.edu (H.C.T.); tblewis@uw.edu (T.B.L.); josborn234@gmail.com (J.O.); jmullins@uw.edu (J.I.M.); 2Washington National Primate Research Center, 1705 NE Pacific Street, Seattle, WA 98195, USA; moats@ohsu.edu (C.M.); Smedley@ohsu.edu (J.S.); 3Stem Cell and Gene Therapy Program, Fred Hutchinson Cancer Research Center, 1100 Fairview Ave. N., Seattle, WA 98109, USA; cwpeters@fhcrc.org (C.W.P.); kjerome@fredhutch.org (K.R.J.); hkiem@fredhutch.org (H.-P.K.); 4Department of Medicine, University of Washington, 1959 NE Pacific Street, Seattle, WA 98195, USA; 5Department of Laboratory Medicine and Pathology, University of Washington, 1959 NE Pacific Street, Seattle, WA 98195, USA; meeili@uw.edu; 6Oregon National Primate Research Center, Oregon Health and Science University, 505 NW 185th Avenue, Beaverton, OR 97006, USA; 7Profectus Biosciences Inc., 6411 Beckley Street, Baltimore, MD 21224, USA; KBagley@orlance.com

**Keywords:** non-human primate (NHP), SHIV, SIV, cART, colon, T helper 17 (Th17), T regulatory, AIDS models, mucosal dysfunction

## Abstract

Selection of a pre-clinical non-human primate (NHP) model is essential when evaluating therapeutic vaccine and treatment strategies for HIV. SIV and SHIV-infected NHPs exhibit a range of viral burdens, pathologies, and responses to combinatorial antiretroviral therapy (cART) regimens and the choice of the NHP model for AIDS could influence outcomes in studies investigating interventions. Previously, in rhesus macaques (RMs) we showed that maintenance of mucosal Th17/Treg homeostasis during SIV infection correlated with a better virological response to cART. Here, in RMs we compared viral kinetics and dysregulation of gut homeostasis, defined by T cell subset disruption, during highly pathogenic SIVΔB670 compared to SHIV-1157ipd3N4 infection. SHIV infection resulted in lower acute viremia and less disruption to gut CD4 T-cell homeostasis. Additionally, 24/24 SHIV-infected versus 10/19 SIV-infected animals had sustained viral suppression <100 copies/mL of plasma after 5 months of cART. Significantly, the more profound viral suppression during cART in a subset of SIV and all SHIV-infected RMs corresponded with less gut immune dysregulation during acute SIV/SHIV infection, defined by maintenance of the Th17/Treg ratio. These results highlight significant differences in viral control during cART and gut dysregulation in NHP AIDS models and suggest that selection of a model may impact the evaluation of candidate therapeutic interventions for HIV treatment and cure strategies.

## 1. Introduction

Since the discovery of HIV in the 1980s, non-human primate (NHP) models of AIDS have been critical for understanding HIV disease pathogenesis and evaluating interventions for HIV prevention and treatment [1]. Infection of NHPs with simian immunodeficiency virus (SIV) and simian-human immunodeficiency virus (SHIV) hybrids have been widely used as models of HIV infection. Vaccines directed towards HIV envelope immunogens cannot be tested using an SIV challenge [2]; therefore, SHIVs were developed by replacing SIV with HIV-1 genes to create a better HIV vaccine challenge model. SIV and SHIV infection recapitulate many aspects of HIV disease, including peripheral CD4 depletion, progression to AIDS, and disruption of gut homeostasis, including depletion of CD4 and T helper 17 (Th17) cells and increased mucosal immune activation [3,4,5]. However, there are significant differences in SIV and SHIV pathogenesis with SIV generally resulting in more accelerated and severe disease [6]. It remains unknown whether SHIV or SIV infection better recapitulates the gut dysfunction during HIV infection and its impact on combinatorial antiretroviral therapy (cART) or immune-based therapeutic interventions.

Reduction in HIV replication by cART has drastically improved the lives of people living with HIV. However, even with cART adherence and viral suppression, HIV reservoirs persist in lymphoid and mucosal tissues [7]. The gastrointestinal tract is a concentrated site of CD4+ T cells, a principal site of early HIV replication and a persistent reservoir during cART in virally suppressed, HIV-infected individuals. Additionally, significant barriers to the success of therapeutic interventions in the gut include mucosal dysregulation and persistent immune activation. Therefore, strategies aiming to augment the decline in mucosal integrity and function may be necessary to achieve a functional cure [8]. We previously showed that rhesus macaques (RMs) infected with SIVΔB670 exhibited highly variable responses to cART [9]. Furthermore, the ability to respond to therapeutic vaccination and control viral rebound after stopping cART correlated with the virological response to cART prior to vaccination [9,10]. We also observed significant loss in mucosal integrity including declines in the Th17/T regulatory (Treg) CD4 T-cell ratio in the colon during acute infection and cART therapy in SIV-infected RMs that correlated with higher immune activation, greater microbial translocation, and increased peripheral immune exhaustion [11]. During HIV infection in humans, gut dysregulation, including impairment of gut Th17 cell responses, is also associated with systemic immune activation and more rapid HIV disease progression [4,12,13].

To determine whether impaired mucosal responses also occurred in a SHIV model, we sought to compare viral replication, cART viral suppression, and gut T cell immune dysregulation during SHIV-1157ipd3N4 infection with SIVΔB670 infection in RM, as previously reported [11]. SIVΔB670 is highly pathogenic and was passaged in vitro to be less pathogenic, but animals still progress to AIDS in less than a year [14,15]. SHIV-1157ipd3N4 is a CCR5-tropic SHIV containing an HIV clade C envelope and was adapted for use in NHPs by passage in vivo and causes progression to AIDS in 2–5 years [16,17]. We demonstrate that both strains support high levels of viral replication in rhesus macaques, with SHIV-1157ipd3N4-infected animals showing lower acute viral burdens and greater virologic control, defined as sustained suppression of viral load below 100 viral RNA copies/mL, during cART. Analysis of mucosal CD4 T-cells during acute infection and following >5 months of cART revealed lower immune dysregulation, defined by an imbalance of Th17 and Treg cells, and lower levels of T-cell activation during cART in SHIV-1157ipd3N4-infected animals indicating overall slower SHIV-related gut disease progression. These results indicate that selection of a pre-clinical NHP model can influence outcomes in the evaluation of HIV treatment and prevention strategies.

## 2. Materials and Methods

### 2.1. Animals, Infection, Antiretroviral Therapy, and Specimen Collection

Male rhesus macaques (RM) were infected intravenously (i.v.) with 100 TCID_50_ SIVΔB670 (n = 19) or 9500 TCID_50_ of SHIV-1157ipd3N4 (n = 24), as previously described [9]. SIV-infected RMs received cART at 6 weeks post-infection (wpi), consisting of subcutaneous (s.c.) daily doses of 9-(2-Phosphonyl-methoxypropyly) adenine (PMPA; Gilead Sciences, Foster City, CA, USA), 2′,3′-dideoxy-5-fluroro-3′-thiacytidine (FTC, Gilead Sciences, Foster City, CA, USA), and oral raltegravir (RAL, Merck & Co., Kenilworth, NJ, USA), as previously described [11]. At 7 wpi, SHIV-infected RMs were given s.c. daily doses of 5.1 mg/kg tenofovir disoproxil fumarate (TDF), 50 mg/kg emtricitabine (FTC), and 2.5 mg/kg dolutegravir (DTG) (Gilead Sciences, Foster City, CA, USA). All animals were housed at the Washington National Primate Research Center (WaNPRC) and all experiments performed on the RMs were approved by the University of Washington’s Institutional Animal Care and Use Committee (IACUC) and were completed in compliance with the U.S. Department of Health and Human Services Guide for the Care and Use of Laboratory Animals and Animal Welfare. Vital signs of the animals were monitored throughout the course of infection for simian AIDS or for adverse reactions associated with the cART regimen under WaNPRC guidelines [11]. Under sedation, blood was collected every 1–4 weeks and colonic pinch biopsies were collected prior to infection (−4 to −2 wpi), during acute infection/prior to initiation of cART (6 wpi), and at 2 weeks (8wpi, SIV only) and 21–22 weeks (28 wpi) after cART initiation, as previously described [11,18]. Lymphocytes were isolated from colon biopsies, as previously described [11].

### 2.2. Plasma Viral Load Quantification

SIVΔB670 plasma viral RNA was evaluated by quantitative real time reverse transcription polymerase chain reaction (RT-PCR) using previously described primers [9] and were determined by the Virology Core at the WaNPRC. SHIV-1157ipd3N4 plasma viral RNA was evaluated by quantitative RT-PCR using published primers and probes on RNA extracted from 1 mL of plasma [19,20]. The limit of detection (LOD) of these assays were 30 (SIV) or 20 (SHIV) copies/mL, therefore viral loads for SHIV-infected RMs were adjusted to LOD 30 copies/mL for each timepoint to be consistent with LOD of SIV-infected RMs.

### 2.3. Intracellular Cytokine Staining

NHP cells isolated from colon biopsies were left unstimulated or stimulated overnight with 10 ng/mL Phorbol 12-Myristate 13-Acetate (PMA; Sigma-Aldrich, St. Louis, MO, USA) and 1 μg/mL Ionomycin (Life Technologies, Carlsbad, CA, USA) in the presence of 1 μg/mL Brefeldin A (Sigma-Aldrich, St. Louis, MO, USA) and CD107a antibody (eBioH4A3, eBioscience^TM^, San Diego, CA, USA), as previously described [11]. Samples were stained as previously described [11]. Briefly, CD4 and CD8 T-cells were identified by CD3 staining following exclusion of doublets and dead cells and selection for CD45+ lymphocytes. Tregs were identified in unstimulated samples by co-expression of CD25 and FoxP3 and Th17 cells in stimulated samples by intracellular expression of IL-17. Polyfunctional Th17 cells were evaluated by Boolean gating for co-expression of IL-22, IFNγ, TNFα, and/or IL-2. Samples were acquired on a LSRII (BD Biosciences, Franklin Lakes, NJ, USA) and analyzed using FlowJo software version 9.9.4 (FlowJo, LLC, Ashland, OR, USA). Staining panels and gating schemes for T-cell subsets are described previously [11].

### 2.4. Blood CD4 and Neutrophil Counts

Complete blood counts (CBC) were used to evaluate the frequencies of neutrophils in the blood and were determined by the University of Washington Department of Laboratory Medicine. Peripheral blood CD4 counts were determined by the University Washington Virology Core, as previously described [21]. Briefly, blood was surface stained, acquired on a FACSCalibur flow cytometer (Becton Dickinson Immunocytometry Systems, Franklin Lakes, NJ, USA) and analyzed with FlowJo software. The absolute number of CD3+CD4+ T-cells was then determined from the CBC.

### 2.5. Statistical Analyses

Kruskal–Wallis test with Dunn’s multiple comparison, Mann–Whitney, or Spearman Rank correlation test statistical analyses were performed and are indicated in the figure legends. A value of *p* ≤ 0.05 was considered statistically significant. All statistical measures were performed using Prism version 8.4.3 (GraphPad, San Diego, CA, USA).

## 3. Results

### 3.1. Lower Viral Burden and Viral Control during cART in SHIV-1157ipd3N4-versus SIVΔB670-Infected Macaques

Rhesus macaques (RMs) were infected intravenously with SHIV-1157ipd3N4 and combination ART (cART), consisting of inhibitors to reverse transcriptase, nucleoside reverse transcriptase, and integrase (TDF, FTC, and DTG) (see Methods), was initiated at 7 weeks post-infection (wpi) (Figure 1). SIVΔB670 infected RMs, as previously reported [11], were used as historic comparators. These animals received a drug regimen consisting of PMPA, FTC, and RAL starting a 6wpi. Optimum challenge doses were given for each viral strain and the best cART regimen available were administered. Blood was collected every 1–4 weeks to measure plasma SHIV/SIV viremia and lymphocytes from colonic gut mucosal tissue biopsies were collected prior to SHIV/SIV infection, during acute SIV infection/prior to initiating cART (6–7 wpi), and following 21–22 weeks on cART (28 wpi) (Figure 1).

Peak SHIV and SIV viremia occurred in all animals 1–2 weeks post-infection (Figure 2a). Peak viremia in the SIV infected animals (7.69 ± 0.67 log_10_) was on average about 1 log higher than viral loads in SHIV-infected animals (6.52 ± 0.47 log_10_). At 6 wpi, prior to cART initiation, mean viral load was also >1.5 log higher in SIV (6.10 ± 1.4 log_10_) versus SHIV-infected (4.11 ± 0.72 log_10_) RMs (Figure 2c) (*p* < 0.0001). Initially a significant decline in plasma viremia occurred in both groups within 4 weeks of cART initiation resulting in 8/19 SIV and 24/24 animals reaching levels of viral replication <100 copies/mL (Figure 2a,b). However, there were significant differences in the response rates to cART, defined by sustained viral control after 5 months of cART, between the SHIV and SIV-infected groups (Figure 2a,b). Specifically, in SHIV-infected RMs, viremia was at or below the limited of detection (<30 copies/mL) in 15/24 (63%) of RMs by 2 weeks post-cART initiation, and in 24/24 (100%) of RMs after 12 weeks of cART (Figure 2b). In contrast, even after 22 weeks (>5 months) on cART only 10/19 (53%) of SIV-infected RMs responded effectively to cART with sustained viremia <100 copies/mL. Of the remaining 9 SIV-infected RMs, 4 had moderate (100–500 copies/mL) and 5 had poor viral control (1000–15,0000 copies/mL) (Figure 2b). Overall, the mean viral burden over >5 months on cART (weeks 9–28, defined as area under the curve, AUC) was just under 1 log higher in SIV-infected RMs (3.58 ± 0.85) when compared to SHIV-infected RMs (2.82 ± 0.23) (*p* < 0.001) (Figure 2d).

### 3.2. SHIV-1157ipd3N4 Infected Macaques Exhibited Less Peripheral CD4 Depletion and Lower Gut Immune Dysfunction and Immune Activation Than SIVΔB670-Infected Rhesus Macaques

We compared SIV- and SHIV-infected macaques to determine the relationship between acute viremia and response to cART and mucosal immune responses. We previously showed that during acute SIV infection, peripheral blood CD4 T-cell counts were initially depleted in the blood [11] but were partially restored by cART, this pattern did not occur during SHIV infection (Figure 3a,b). In contrast, CD4 T-cells declined in the colonic mucosa in both groups during acute infection and cART partially restored these cell levels consistently between the two infection groups (Figure 3c,d, Appendix A Appendix A). We also looked at the CD4/CD8 ratio, a biomarker of HIV disease progression [22,23], in the blood and colon. Significant declines in the blood CD4/CD8 ratio occurred in SIV, but not SHIV animals (Appendix A Appendix A). However, there was a marked decrease in the colonic CD4/CD8 ratio during acute infection, with partial restoration during cART (Appendix A Appendix A). Neutropenia during acute infection is also a biomarker of disease progression [24], however neither infection resulted in notable declines of neutrophils in the blood (Appendix A Appendix A).

Lower peripheral Th17/Treg ratios during chronic HIV infection are associated with more rapid progression to AIDS [12]. Likewise, we previously reported that declines in mucosal Th17/Treg ratios corresponded with increased immune activation and peripheral immune exhaustion in SIV-infected RMs [11]. We therefore compared the impact of SIV vs. SHIV infection on T-cell subsets known to be important for gut homeostasis during HIV infection. To account for individual animal variability that could be introduced by differences in lymphoid tissues sampled during the colon biopsy and to measure changes in immune responses following infection, we report immune responses as a Log_10_ fold-change from baseline. Although overall CD4 depletion and restoration kinetics in the colon were similar in SIV and SHIV-infected animals (Figure 3), the magnitude and impact on certain CD4 cell types were distinct between the two infections. During the acute phases of infection (weeks 0–6), Treg cells expanded and Th17 cells declined in SIV-infected animals, resulting in a significantly greater decline in the Th17/Treg ratio during SIV when compared to SHIV infection (Figure 4a–e, Appendix A Appendix A), indicating greater gut dysregulation in the SIV-infected RMs. After 5 months of cART, Th17 cells remained depleted in both groups and levels were comparable between the groups. However, the Treg expansion remained significantly higher in the SIV-infected RMs (*p* = 0.0027) and thus, overall, SIV-infected RMs sustained significantly greater declines in Th17/Treg ratios during cART when compared to SHIV-infected RMs (Figure 4f). Interestingly, although the decline in Th17 cells in the colon during cART was similar between SIV- and SHIV-infected RMs, the loss of Th17 polyfunctionality was greater in SIV-infected RMs (Figure 4f, Appendix A Appendix A). This suggests that different inflammatory environments elicited during SIV vs. SHIV infection, likely related to the magnitude of persistent immune activation, may have promoted the differences we observed in the expansion and maintenance of Treg cells between these groups during cART. Increases in CD4 and CD8 T-cell immune activation (%Ki67^+^) were similar between SIV- and SHIV-infected RMs during acute infection (Figure 5a–d). Immune activation during cART was dramatically reduced in both groups but was lower in SHIV-infected animals (Figure 5a–d, Appendix A Appendix A). Collectively, SHIV-1157ipd3N4 infected animals, consistent with lower acute viral burdens and better viral suppression on cART, had less CD4 depletion, disruption of gut homeostasis, and immune activation during acute infection and during cART compared to SIVΔB670-infected animals.

### 3.3. Lower Acute Gut Immune Dysfunction in SIV/SHIV-Infected Macaques Is Associated with Greater Viral Control by cART

We next sought to understand the role of SIV/SHIV gut pathogenesis, in particular CD4 cell subset dysregulation, on viral control during cART. To test this, we compared the fold change in colonic CD4 T-cell populations during acute SIV/SHIV infection to the levels of viral burden following cART. This analysis revealed that maintenance or increases in Th17, Th17 polyfunctionality or the Th17/Treg ratio was associated with a lower viral burden during the period of cART (Figure 6a,b,d). In contrast, expansion of T regulatory cells in the colon during acute infection was associated with greater viral burden during (Figure 6c). These data indicate that less gut immune T cell dysregulation during acute infection is associated with lower viral burden during cART.

## 4. Discussion

The success of NHP AIDS models has led to increased demand in NHPs in recent years [25], but appropriate selection of a pre-clinical SIV/SHIV NHP model is critical when evaluating HIV interventions. In these studies, we report less variability in viral control during cART in SHIV-1157ipd3N4 versus SIVΔB670-infected RMs. Specifically, 9/19 SIV-infected animals did not exhibit sustained virologic control during cART compared to SHIV-infected animals which maintained low/undetectable levels of viral replication within a few weeks of starting cART. During treated human HIV infection viral suppression is regularly achieved within 12–24 weeks [26] and is highly consistent with the responses in SHIV-infected animals reported here. Historically, control of SIVΔB670 replication by antiviral drugs has been highly variable, with some animals unable to control viral replication despite 9 months of treatment [9]. While we cannot rule out that longer treatment could result in undetectable levels of plasma viremia, our results indicate that greater restoration of mucosal T-cell functions occurs in animals that exhibit more immediate control of viremia after initiating cART. These studies were limited to evaluation of viral burden in the blood. Studies in NHP have shown a relationship between plasma and tissue levels of virus replication [27,28] and decreased penetrance of antiretrovirals to mucosal sites [29], so it is likely that animals with high plasma viremia also had greater viral burden at mucosal sites. Future studies are needed to evaluate the viral burden in mucosal compartments to determine the relationship between local viral replication and mucosal CD4 T-cells more conclusively. Antiretroviral drugs designed to inhibit HIV replication can be less efficient in suppressing SIV viral replication and can often take months to achieve durable viral suppression and our findings are consistent with previous reports [30,31]. Although the classes of cART regimens were similar, studies in humans have revealed higher interpatient pharmacokinetic variability with RAL, that was used in ourSIV infected RMs, and less variability with DTG, that was given to our SHIV infected RMs [32]. This may explain the increased variability in viral control we see in the SIV-infected RMs. In addition, all three drugs given to the SHIV-infected animals were injectable, whereas one drug (RAL) was delivered orally in the SIV-infected animals; this may have influenced relative cART “adherence” and effectiveness of this regimen in the SIV-infected animals. Alternatively, if the SIV-infected animals were given the same ART regimen as the SHIV-infected animals (e.g., TDF, FTC, DTG) greater viral suppression may have occurred. This specific drug regimen was given to 3 SIVmac239X infected rhesus macaques 2 weeks after infection, yet stable viral suppression was also not achieved until 12–20 weeks following cART [33]. These data suggest that even if the SIVΔB670-infected RMs had been given a cART regimen of TDF, FTC, DTG, viral suppression may have been incomplete or taken longer than during SHIV infection, thus allowing a greater period to disrupt gut immunostasis.

Sustained virologic control during cART, primarily in SHIV-1157ipd3N4-infected RMs, corresponded with less dysregulation of the Th17/Treg ratio during acute infection. These results highlight differences in viral suppression during cART responsiveness in NHP models during cART with 100% of SHIV-1157ipd3N4 and only 53% of SIVΔB670-infected RMs reaching virologic control on cART. Furthermore, better virologic control in response to cART corresponded to less mucosal immune dysfunction, including higher Th17/Treg ratios and lower gut immune activation in SHIV than in SIV-infected animals. These results demonstrate that the SIV and SHIV NHP AIDS models recapitulate aspects of mucosal immune dysfunction found during HIV infection, with signs of increased mucosal pathogenesis in SIVΔB670-infected RMs. Additionally, we found that increased pathogenesis during acute infection leads to greater disruption of mucosal CD4 T-cells and corresponds to poorer virologic control during cART. However, several differences in the experimental design, including virus strain, inoculation dose, time of cART initiation and drug regimen, etc. can influence the mucosal environment, and therefore, differences between the SHIV and SIV models that we report here. Thus, the level of viral pathogenesis not only in the periphery, but also at mucosal sites, needs to be considered when determining whether to use SHIV or SIV NHP models in HIV vaccine studies.

Early cART treatment during HIV infection has been shown to preserve CD4 and Th17 cells in the gut, but has been less successful in reversing persistent gut inflammation, reducing chronic immune activation, or eliminating viral reservoirs [34,35]. During treated human HIV infection, early cART initiation was found to restore Th17 cell numbers, but not Th17 polyfunctionality and demonstrates how early stages of the infection can have detrimental long-term influences on mucosal T-cell compartments [36]. Consistent with these findings, we found that mucosal CD4 and Th17 restoration during cART was similar in both infection models, restoration of Th17 polyfunctionality was greater in SHIV infected RMs. Therefore, the SHIV model reported here may better recapitulate aspects of gut mucosal dysfunction observed in cART-treated people living with HIV. This is likely due to the lower viral replication and more efficient suppression of viral replication on cART in the SHIV model. Although cART improves some aspects of HIV infection and disease progression, the findings reported here, including chronic immune activation and dysregulation of Th17 and Treg cells, suggest that strategies to improve gut mucosal homeostasis may increase the effectiveness of immunotherapies aimed to treat or cure HIV infection. Furthermore, our results provide strong evidence that both the SIV and SHIV macaque models of AIDS recapitulate declines in mucosal T-cell function and immune activation, but that the SHIV infection model may more closely mimic this relationship in treated people living with HIV.

Structural barriers and gaps in the HIV care continuum have led to many people living with untreated HIV. As of 2020, 27% of people with a known HIV status did not have access to cART and of those receiving treatment 44% were not virally suppressed [37]. Failure to achieve long-term HIV viral suppression during cART, due to noncompliance, drug resistance, or other factors, is associated with increased risk of AIDS- and non-AIDS associated co-morbidities [38] and could ultimately influence the efficacy of HIV intervention strategies. Spontaneous viral remission occurs in untreated individuals [39] or post-treatment controllers after analytic treatment interruption [40] and can lead to overestimations in the effectiveness of therapies. Thus, the SIV model, which exhibited greater immune dysregulation consistent with suboptimal cART regimens, may better recapitulate individuals with less accessibility or poorer adherence to cART. Variability of virologic control during cART in pre-clinical NHP AIDS models may impact the relative evaluation of candidate therapeutic interventions for HIV cure in humans and selection of an appropriate NHP model that represents the target population for a given therapeutic is critical. The SHIV macaque model may be better for identifying initial vaccine candidates, but it may be necessary, when possible, to switch to a SIV model, that has greater variability in cART viral control and gut immune dysfunction, to model aspects of real-world scenarios of cART nonadherence or unsuppressed HIV replication.

## Figures and Tables

**Figure 1 viruses-13-01609-f001:**
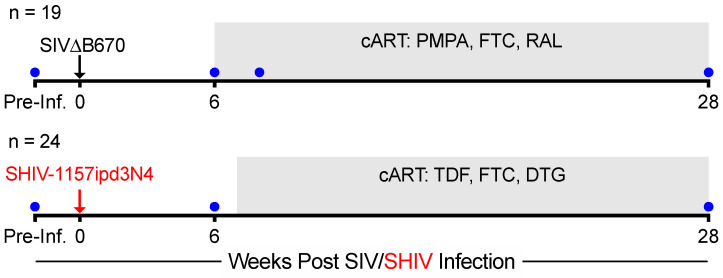
Study Design. Rhesus macaques were infected intravenously (i.v.) with SIVΔB670 (*n* = 19) or SHIV-1157ipd3N4 (*n* = 24). Animals received daily cART starting 6 (SIV) or 7 (SHIV) weeks post-infection and consisted of injectable PMPA, FTC and oral RAL (SIV) or injectable TDF, FTC, and DTG (SHIV). Blood was collected every 1–4 weeks. Colon biopsies (blue circles) were collected pre-infection (SIV: −4 to −2 wpi; SHIV: −1 to −2 wpi), during acute infection/prior to initiation of cART (6 wpi), 2 weeks after cART initiation (8 wpi, SIV only), and at 21–22 weeks (28 wpi) after cART initiation.

**Figure 2 viruses-13-01609-f002:**
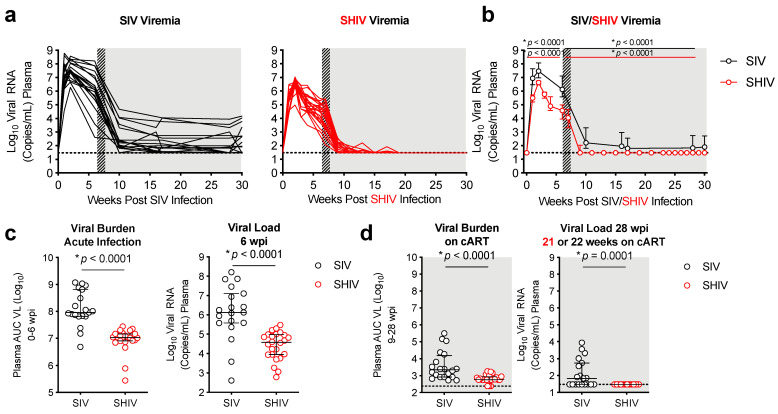
SHIV-infected macaques have lower viral burden and better viral control during cART therapy. (**a**,**b**) Plasma SIV or SHIV (viral RNA levels at each timepoint were measured by RT-PCR (SIV) or qRT-PCR (SHIV). The dotted line indicates the limit of detection (30 copies/mL of plasma). Hashed and shaded regions indicate periods of cART treatment for SIV RMs and all animals, respectively. Lines indicate individual animals (**a**) or medians with interquartile ranges (**b**). (**b**) Kruskal–Wallis test, adjusted *p*-values are given. (**c**) Area under the curve (AUC) of plasma viremia during acute infection (0–6 wpi) (left panel) and viral load at 6 wpi (right panel). (**d**) AUC of plasma viremia during cART treatment (9–28 wpi) (left panel) and viral load 21–22 weeks on cART (right panel). (**c**,**d**) Individual RMs are indicated by circles: SIV (black open circles), SHIV (red open circles). Medians with interquartile ranges are indicated. *p*-values were determined using a Mann–Whitney Test, with * *p*-values ≤ 0.05 considered significant.

**Figure 3 viruses-13-01609-f003:**
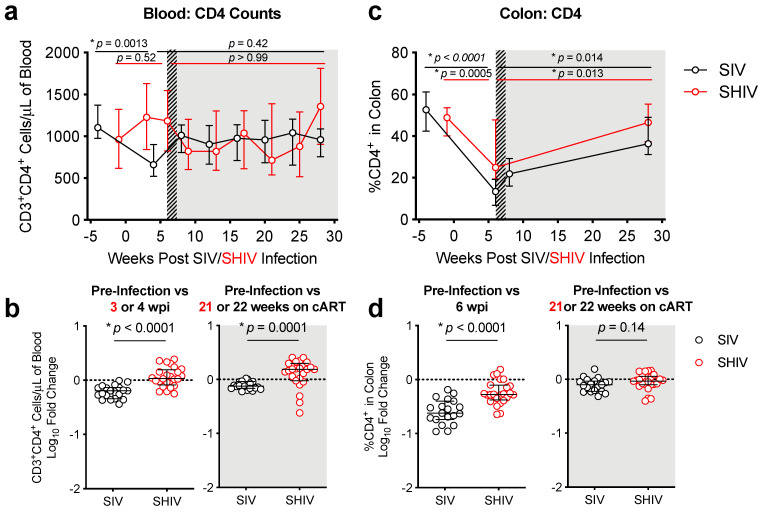
Peripheral and mucosal CD4 depletion in SIV-infected, but only mucosal depletion in SHIV-infected macaques. (**a**) CD4^+^ T-cells in the blood were quantified from the complete blood count (CBC) following flow cytometry analysis. (**c**) The frequency of CD4^+^ T-cells within the CD45^+^ T-cell subset from colon biopsies was determined by flow cytometry. (**a**,**c**) Hashed and shaded regions indicate periods of cART treatment for SIV RMs and all animals, respectively. Lines indicate medians with interquartile ranges. Kruskal–Wallis test, adjusted *p*-values are given. (**b**,**d**) Log_10_-fold change in CD4 T-cells between pre-infection and acute infection at 3 (SHIV) or 4 (SIV) weeks post-infection (wpi) or after 21 (SHIV) or 22 (SIV) weeks (>5 months) on cART in the blood (**b**) or colon (**d**) between SIV (black open circles) and SHIV (red open circles). Each dot represents an individual RM with medians and interquartile ranges indicated. The dotted line indicates no change (value of 1). *p*-values were determined using a Mann–Whitney test, with * *p*-values ≤ 0.05 considered significant.

**Figure 4 viruses-13-01609-f004:**
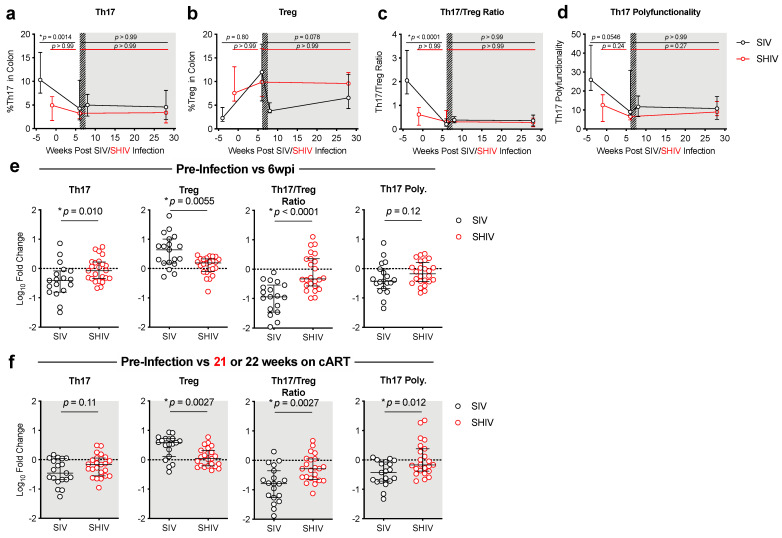
Higher viral burden during acute SIV infection and during cART is associated with greater gut dysfunction. (**a**,**b**) The percentage of Th17 (IL-17^+^) (**a**), stimulated with PMA and ionomycin, or unstimulated Tregs (FoxP3^+^CD25^+^) (**b**) of CD4^+^ T-cells in the colon was determined by flow cytometry (as described in methods). (c) The Th17/Treg ratio was determined by dividing the percentage of IL-17^+^ of CD4^+^ cells by the percentage of FoxP3^+^CD25^+^ of CD4^+^ cells. (**d**) Polyfunctional Th17 (Th17 Poly.) cells were identified as the percentage IL-17^+^ cells that produced at least two additional cytokines (IL-22, IFNγ, TNFα, or IL-2) as determined by Boolean gating following in vitro stimulation with PMA and ionomycin. (**a**–**d**) Hashed and shaded regions indicate periods of cART treatment for SIV RMs and all animals, respectively. Lines indicate medians with interquartile ranges. Kruskal–Wallis test, adjusted *p*-values are given. (**e**–**f**) Log_10_-fold change in cells in the colon between pre-infection and acute infection or 21–22 weeks (>5 months) on cART between SIV (black open circles) and SHIV (red open circles). Each dot represents an individual RM with medians and interquartile ranges indicated. The dotted line indicates no change (value of 1). *p*-values were determined using a Mann–Whitney test, with * *p*-values ≤ 0.05 considered significant.

**Figure 5 viruses-13-01609-f005:**
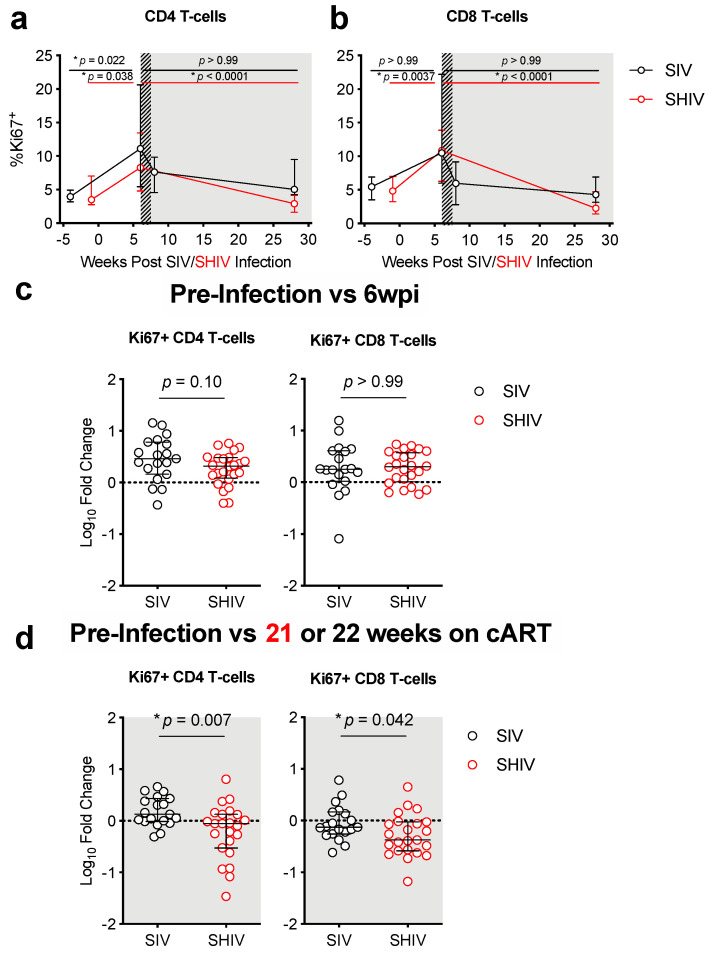
Incomplete viral suppression during cART in SIV-infected macaques is associated with greater gut immune activation. (**a**,**b**) Immune activation was measured as the percentage Ki67^+^ of CD4^+^ (**a**) or CD8^+^ (**b**) T-cells in the colon as determined by flow cytometry (as described in methods). (**a**,**b**) Hashed and shaded regions indicate periods of cART treatment for SIV RMs and all animals, respectively. Lines indicate medians with interquartile ranges. Kruskal–Wallis test, adjusted *p*-values are given. (**c**,**d**) Log_10_-fold change in immune activation in the colon between pre-infection and acute infection or 21–22 weeks (>5 months) on cART between SIV (black open circles) and SHIV (red open circles). Each dot represents an individual RM with medians and interquartile ranges indicated. The dotted line indicates no change (value of 1). *p*-values were determined using a Mann–Whitney test, with * *p*-values ≤ 0.05 considered significant.

**Figure 6 viruses-13-01609-f006:**
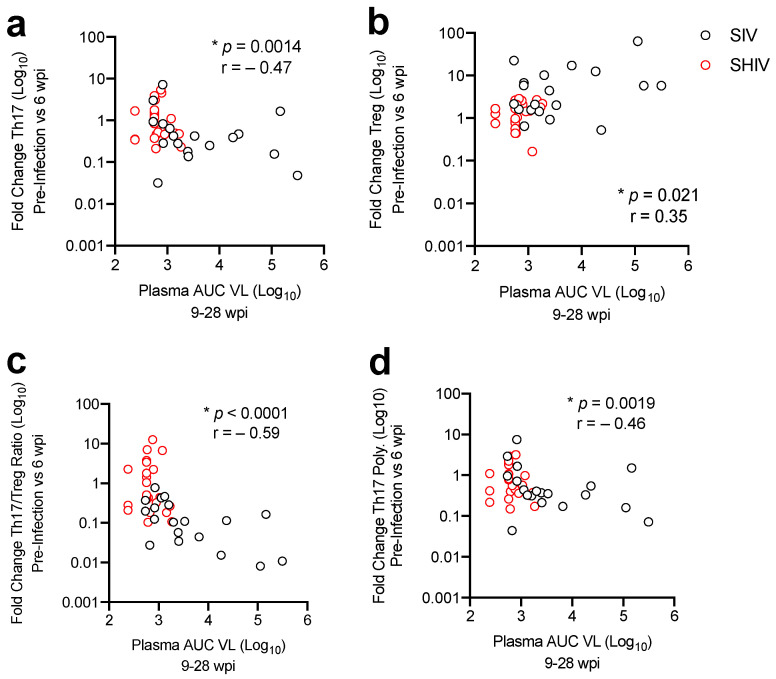
Less dysregulation of mucosal CD4 T-cells is associated with greater viral control during cART. Correlations between log10-fold change in colonic (**a**) Th17 and (**b**) Treg cell frequencies, (**c**) the Th17/ratio, and (**d**) Th17 polyfunctionality (Th17 Poly.) during acute infection (weeks 0–6) versus AUC plasma viremia during cART (weeks 9–28). Spearman’s rank correlation is shown, with * *p*-values ≤0.05 considered significant.

## Data Availability

The data that support the findings of this study are available from the corresponding author upon reasonable request.

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
