# Peer review of "A Gut Reaction to SIV and SHIV Infection: Lower Dysregulation of Mucosal T Cells during Acute Infection Is Associated with Greater Viral Suppression during cART"

_viruses, 2021, doi:10.3390/v13081609_

Round 1

Reviewer 1 Report

My major concern with this paper is the way the the previously published SIV work is described. It would be much more appropriate and easier to understand if you simply write that the SIV work was done previously and here, you are comparing that published work with a new study using SHIV. As it reads now, it is clear that you did some Th17 work in a previous paper, but it is NOT clear that the actual same data set are in both papers. The figures are clearly the same with the addition of the SHIV on top. This also clarifies why the drug regimen is different and why time to ART changes. The entire paper needs to be re-worded so that it is clear that the SIV work is published and the conclusions from that study. And that here, you are simply comparing that study with a new one with SHIV. During the comparison sentences, they need to read like, we previously showed X and Y in SIV, and here we found similar or different results. I am not suggesting that you were hiding the previous work, it is clearly listed in abstract and introduction, but that the text reads like there were two studies set up to test this.

"Comparable cART regimens" is not really accurate. Again tone is off. If you had used the SIV animals as historic comparators, then stated that this new study with SHIV used the best available cART (TDF, FTC, and DTG) that would be the same information but described more accurately. There are going to be differences, and that is fine, but the rationale for those differences is much more palatable if the readers understand the context. This is critical because the SIV-infected animals were the ones with the suboptimal cART--half the animals or more had detectable viremia. 

Reviewer 2 Report

In the report by O’Connor and colleagues, the authors compare SIV/SHIV viral kinetics and dysregulation of gut homeostasis upon infection of rhesus macaques with two differently, widely used viruses, SIVdeltaB670 and SHIV-1157ipd3N4. Comparisons are made during acute infection, virus control upon cART and during therapy up to 28 weeks post-infection. The authors find that SHIV infection resulted in lower acute viremia, less disruption to gut CD4 T-cell homeostasis (defined by the Th17/Treg ratio), and better sustained viral suppression <100 copies/ml of plasma after 5 months of cART. They conclude by highlighting that NHP AIDS model selection for therapeutic intervention studies should be done with caution due to the differences in the diverse models available.

  1. The most important criticism to this study is the fact that the authors start with the assumption that the experiments performed with the two models are completely comparable, and they should assure that this is the case. The infections have been performed with two different viruses (which is the basis to the comparison), but also with different TCID50 infectious doses, distinct cART regimens (including oral vs. i.v. administration), and distinct assays for viral load quantification (including different primers and assay platforms) in each group. Although the authors have discussed a little bit the potential bias related to the differing cART regimens used in the two animal groups, the most important thing in my view is to assure that infectious dose inputs were comparable, otherwise the whole interpretation of the results obtained may be compromised, because they may not rely on the virus, but rather on the infectious dose used.
  2. Authors should provide more details on the intracellular cytokine staining targets used in section 2.3 of Methods, instead of just citing a reference for that.
  3. With respect to the statistical analyses performed, why the authors did not try to adjust the p-values for multiple comparisons throughout all the experiments?

Cosmetic issues:

Line 236, legend to Figure 2: the last word of the line should read “SHIV”, not “SIV”.

Reviewer 3 Report

In this paper the authors are comparing SIV and SHIV infected rhesus macaques and evaluating the effect of cART introduction in acute phase on systemic viral suppression, circulating CD4+ population and mucosal CD4, CD8, activation (KI67), T reg and Th17 populations. the authors are raising an extremely important point based on the few differences they observed in those models which is that the selection of a model while designing a study might impact the results like in the evaluation of therapeutic interventions for HIV cure. 

There is some suggestions that I would like to make and questions that I would like to raise with the goal to improve this manuscript and it's message. 

1- I would like to suggest to review the title to make it more specific to the authors findings. the authors are using in the tittle and across the manuscript the words "mucosa or gut immune dysregulation", this to me, imply a much broader look at multiple immune parameters in the GI- In this manuscript, the authors are focusing on the CD4+ T cells, CD8+ cells, KI67, Treg and Th17 cells/markers. To me this doesn't constitute a complete analysis of the immune environment in the GI. Please consider using a title that reflects more what you are demonstrating maybe based on your key words. 

2- Regarding the study design, the animals in this study didn't receive the exact same ART regiment, even if the general drugs are really similar (2NRTI+1INSTI), each of the INSTIs have unique pharmacokinetic / pharmacodynamic properties, influencing their role in clinical use in specific subsets of patients/animals. I think it would be best to add comments in discussion about this, there is also a nice complete review comparing INSTI (27317415). 

3- In the results, part1 , the authors are talking about a better responsiveness to cART in animals infected with SHIV. How can the authors claim that the SHIV infected respond better to the treatment since those animals present 2 logs or more of difference in blood VL to start with? 

4- Can the authors explain why the detection of CD4+T cells in the blood and in the colon was not performed using the same assay? why not using Flow approach for both? 

5- For the Bx samples it looks like the SIV animals went under an extra Bx right after cART initiation, why this wasn't the case for SHIV animals? different protocols?

6- While looking at the Th17 and Treg population in the colon bx, there was some important disparity within the groups, how do the authors explain these differences? This might reveal a Bx sampling limitation, can reflect presence or absence of lymphoid aggregates between pinches, or more or less local infection especially since SIV are not fully suppressed, or local damage with bacterial translocation and more local activation... Did the authors look at the population of CD20 cells to know if they were sampling some lymphoid aggregates? 

7- An essential results is missing in this manuscript, the tissue VL in colon Bx or presence of virus in tissue using in situ approach. This data would be essential to understand the differences between the Th17/Treg ratio but also the level of activation and depletion of CD4+T cells. 

8- Did the authors looked at other parameters to better identify a gut immune dysregulation like claimed across the manuscript? Like sCD14, Neutrophils level in blood and GI, T cell exhaustion (TIGIT), Mx1,  bacteria flora... 

9- Can the authors discuss of the benefit of having a longer study to wait until all animals reach undetectable blood VL?

General comments:

  • Line 146: author do not talk about integrate inhibitors in the regiment of ART administrated.
  • In general the graphs are redundant between longitudinal curves and fold change, maybe the authors can place a line on the Y axes across the graph to represent average base line and just put the fold change number between line and last time point. 
  • Please use same scale for Y axes for easier comparaison.
  • Figure 6 title should contain Th17/Treg population and not the general broad appellation of "gut dysregulation".
  • line 319, the ref are studies performed in RM while authors are talking of HIV infected patients.

Good luck!

Round 2

Reviewer 2 Report

The authors have addressed all my criticisms in the revised version of the manuscript, and provided sufficient information. Therefore, I think the ms is now suitable for acceptance in Viruses.

Reviewer 3 Report

Thanks for taking the time to answer clearly and adequately to all comments, questions and suggestions raised. The manuscript looks good and I highly support its publication.